# The Anti-Tumoral Potential of Phosphonate Analog of Sulforaphane in Zebrafish Xenograft Model

**DOI:** 10.3390/cells10113219

**Published:** 2021-11-18

**Authors:** Magdalena Rudzinska-Radecka, Łukasz Janczewski, Anna Gajda, Marlena Godlewska, Malgorzata Chmielewska-Krzesinska, Krzysztof Wasowicz, Piotr Podlasz

**Affiliations:** 1Foundation of Research and Science Development, Rydygiera 8, 01-793 Warsaw, Poland; magdda.rudzinska@gmail.com; 2Institute of Molecular Medicine, Sechenov First Moscow State Medical University, 119991 Moscow, Russia; 3Institute of Organic Chemistry, Faculty of Chemistry, Lodz University of Technology, Zeromskiego 116, 90-924 Lodz, Poland; lukasz.janczewski@p.lodz.pl (Ł.J.); anna.gajda@p.lodz.pl (A.G.); 4Department of Biochemistry and Molecular Biology, Centre of Postgraduate Medical Education, Marymoncka 99/103, 01-813 Warsaw, Poland; marlena.godlewska@cmkp.edu.pl; 5Department of Pathophysiology, Forensic Veterinary Medicine and Administration, Faculty of Veterinary Medicine, University of Warmia and Mazury in Olsztyn, 10-719 Olsztyn, Poland; malgorzata.chmielewska@uwm.edu.pl (M.C.-K.); wasowicz@uwm.edu.pl (K.W.)

**Keywords:** zebrafish, isothiocyanates, glioblastoma, cervical cancer, breast cancer, toxicology, zebrafish xenograft model

## Abstract

Isothiocyanates (ITCs) show strong activity against numerous human tumors. Five structurally diverse ITCs were tested in vivo using the zebrafish embryos 6 and 48 h post-fertilization (hpf). The survival rate, hatching time, and gross morphological changes were assessed 24, 48, and 72 h after treatment with all compounds in various doses (1–10 µM). As a result, we selected a phosphonate analog of sulforaphane (P-ITC; 1–3 µM) as a non-toxic treatment for zebrafish embryos, both 6 and 48 hpf. Furthermore, the in vivo anti-cancerogenic studies with selected 3 µM P-ITC were performed using a set of cell lines derived from the brain (U87), cervical (HeLa), and breast (MDA-MB-231) tumors. For the experiment, cells were labeled using red fluorescence dye Dil (1,1′-Dioctadecyl-3,3,3′,3′-Tetramethylindocarbocyanine, 10 μg/mL) and injected into the hindbrain ventricle, yolk sac region and Cuvier duct of zebrafish embryos. The tumor size measurement after 48 h of treatment demonstrated the significant inhibition of cancer cell growth in all tested cases by P-ITC compared to the non-treated controls. Our studies provided evidence for P-ITC anti-cancerogenic properties with versatile activity against different cancer types. Additionally, P-ITC demonstrated the safety of use in the living organism at various stages of embryogenesis.

## 1. Introduction

Isothiocyanates (ITCs; sulfur-containing substances) are naturally occurring small molecules formed from glucosinolate precursors in cruciferous vegetables, including cabbage, kale, broccoli, cauliflower, Brussels sprouts and kohlrabi [1]. It was already shown that both natural and synthetic ITCs display anti-carcinogenic activity by affecting multiple mechanisms, including apoptosis, metastasis, vascularization, cell cycle, oxidative stress and epigenetic mechanisms [2,3] (Figure 1).

The analysis has shown that ITCs chemopreventive activity is exerted due to favorable modification of phase I and phase II enzymes activity and carcinogen metabolism. This reaction increases carcinogen secretion or detoxification that consequently decreases carcinogen-DNA interactions [5]. In this scenario, a variety of ITCs can prevent development of many tumor types, such as murine osteosarcoma [6], human mammary gland [7,8], esophagus [9], liver [10], colon [11], and bladder [12] carcinomas. Many studies presented structural diversity of ITCs, and their synthetic analogs have been successfully tested [8,13]. ITCs are considered to be safe without serious adverse effects in humans [14]; moreover anti-oxidant [15], anti-inflammatory [16], anti-microbial [17], neuroprotective [18] and cardioprotective [19] effects were reported.

Recent progress in pharmaceutical screening with zebrafish (*Danio rerio*) and its use as a xenograft model of human tumors has shown that zebrafish is a reliable organism for evaluating safety of new treatments and cancer xenotransplantation studies [20]. The transparent embryos of zebrafish, which develop outside the maternal organism with most organ primordia being formed 24 h after fertilization (hpf), have made zebrafish the ideal subject in toxicity tests [21]. Additionally, a series of adverse morphological phenotypes and behavioral endpoints are used to predict the mechanism of action for treatment perniciousness [22]. In the context of xenotransplantation, the embryonic zebrafish xenografts of human tumors can be a beneficiary laboratory model that can be used to understand cancer biology and as a reliable alternative for mice in the studies of drugs that target tumor progression [23,24].

Five structurally different ITCs (LJ-NCS-1–5) were previously designed and tested in vitro using cell lines derived from human colon, lung, mammary gland and uterus carcinomas and in vivo by a murine mammary gland carcinoma model. All compounds exhibited high anti-proliferative activity, with the potential to induce G2/M cell cycle arrest and apoptosis.

Here, we aimed to use zebrafish embryos as an alternative vertebrate model to identify non-toxic drug candidates at the different stages of embryogenesis. As a result, we selected the 3 µM phosphonate analog of sulforaphane (LJ-NCS-2; P-ITC) as the safest treatment without visible teratogenic effects. Next, we tested the anti-cancerogenic properties of P-ITC for cell lines derived from the brain (U87), cervical (HeLa) and breast (MDA-MB-231) carcinomas, which were injected into the hindbrain ventricle, yolk sac region and Cuvier duct of zebrafish embryos, respectively. The tumor size measurement after P-ITC treatment demonstrated the significant inhibition of cancer cell growth in all studied cases.

To our knowledge, the anti-tumoral potential of P-ITC was never investigated using zebrafish embryo xenografts.

## 2. Materials and Methods

### 2.1. Chemistry

The tested ITCs are presented in Table 1.

ITCs LJ-NCS-1–5 are known compounds and were previously synthesized using three described methods.

Isothiocyanates LJ-NCS-1, LJ-NCS-4, and LJ-NCS-5 were obtained in one-pot, two-step microwave-assisted synthesis. In the first step, primary amines LJ-1, LJ-4, and LJ-5 in the presence of carbon disulfide (CS_2_) and triethylamine (Et_3_N) were transformed in normal reaction at room temperature and in 5 min time to dithiocarbamates LJ-D-1, LJ-D-4, and LJ-D-5. They were then converted to final ITCs with good and excellent yields (61–87%) in the second step in a microwave reactor (20 min, 90 °C) without any desulfurating reagent (Figure 2) [28].

ITC LJ-NCS-3 was also synthesized in a one-pot, two-step microwave-assisted synthesis using CS2, Et3N and hydrochloride LJ-3 as substrate under changed microwave treatment. In the first step, intermediate dithiocarbamate LJ-D-3 was obtained under the same conditions as LJ-D-1, LJ-D-4, and LJ-D-5 (Figure 2); however, the second step–desulfurization of LJ-D-3-was performed using microwave-assisted synthesis in a shorter time and the same temperature (3 min, 90 °C), and additionally in the presence of desulfurating agent–DMT/NMM/TsO^−^. LJ-NCS-3 was isolated with efficiency of 75% (Figure 3) [26].

The last ITC LJ-NCS-2 was obtained using another method based on the tandem Staudinger/aza–Wittig reaction and using azide LJ-2 as a substrate. In this synthesis diethyl 6-azidohexylphosphonate (LJ-2) [29] was converted with high yields (75%) in the corresponding diethyl 6-(isothiocyanate)hexylphosphonate (LJ-NCS-2) in a one-pot, two-step reaction with triphenylphosphine (PPh_3_) and CS_2_ (Figure 4) [8].

### 2.2. Zebrafish Maintenance

Transgenic zebrafish Tg(fli1:EGFP)y1 strain (a transparent zebrafish kind that expresses enhanced GFP in the entire vasculature under the control of the fli1 promoter, and thus enables the visualization of vascular defects in live zebrafish embryos) was kept at 27 °C in aquaria with 14/10h day/night light cycle (14 hours of light/10 hours of darkness) The embryos were raised at 28,5 °C in E3 medium (5 mM NaCl. 0.17 mM KCl. 0.33 mM CaCl_2_. 0.33 mM MgSO_4_) until the desired developmental stages [24]. Embryos at 6 and 48 hpf were treated 24, 48 and 72 h with ITCs in doses: 1–10 µM.

### 2.3. Cell Culture

The U87 (human glioblastoma), HeLa (cervical tumor) and MDA-MB-231 (triple-negative breast cancer) were obtained from American Type Culture Collection (ATCC).

Cells were cultured in Dulbecco’s modified Eagle’s medium (DMEM; Life Technologies, Gibco, NY, USA) or RPMI-1640 medium (Life Technologies, Carlsbad, CA, USA) supplemented with 10% fetal bovine serum (FBS; Roche, Mannheim, Germany) at 37 °C in a humidified 95% air and 5% CO_2_ atmosphere. All cell lines were checked using PCR assay for mycoplasma and were free of contamination.

### 2.4. Embryotoxic Test of Isothiocyanates in Zebrafish

For embryotoxicity evaluation, 50 healthy embryos of AB wild-type strain (6 and 48 hpf) were used. Different concentrations of ITCs (110 µM) suspended in 0.05% of DMSO in E3 medium were added to the well plate and incubated at 28.5 °C for 48 h. The control was treated with 0.05% of DMSO in E3. The zebrafish embryos were evaluated for malformations after 24, 48 and 72 h using stereomicroscope (SteREO Discovery.V8, Carl Zeiss, Oberkochen, Germany) to evaluate the embryotoxicity of the substances. In this scenario, morphological phenotypes, including malformed yolk sacs, tail malformations, delayed development and impaired motor activity, were observed. The motility of the embryos was recorded by microscopic observation; in addition, for experiment with treatment of 6 hpf embryos after 24 h treatment, the 5 min. videos were recorded and analyzed by DanioScope software (Noldus, the Netherlands). The hatching rate was monitored at 72 hpf stage of development for experiment with treatment of embryos from 6 hpf.

### 2.5. Microinjection of Human Tumor Cells into Zebrafish Embryos

The U87, HeLa and MDA-MB-231 cells were washed, trypsinized and re-suspended in PBS. Then the cells were stained with lipophilic red fluorescence dye Dil (1,1′-Dioctadecyl-3,3,3′,3′-Tetramethylindocarbocyanine Perchlorate; 10 μg/mL; ThermoFisher, Waltman, MA, USA) for 20 min. Furthermore, cells were washed three times in PBS and re-suspended in PBS with 0.5 mM EDTA; the technique was previously described [30]. Before the experiment, part of non-dechorionated embryos was entirely immersed in embryo medium, transferred with a glass pipette, and maintained in agarose-coated (1% in embryo medium) plastic dishes. Then, the medium was carefully poured out and the plastic dish was slowly filled with fresh embryo medium. The steps were repeated three times altogether. The mild agitation facilitated the removal of the chorions.

For microinjection of tumor cells, the Tg(fli1:EGFP)y1 zebrafish embryos 48 hpf were anesthetized with 0.02% tricaine (MS-222; Sigma-Aldrich, Saint Louis, MO, USA). A total of 500 cells (5 nL) were injected into the (U-87) hindbrain ventricle as described [31], (HeLa) middle of the embryonic yolk sac region [32] and (MDA-MB-231) into Cuvier duct through which cells migrated to the caudal hematopoietic tissue [33]. All embryos were separately seeded in a 48-well plate for a detailed monitoring of each individual.

### 2.6. Xenograft Exposure Assay

24 h after injection the viable embryos with fluorescent cells were screened with a fluorescence microscope and selected for further 48 h treatment. The embryos were incubated at 35 °C with 3 µM P-ITC or without P-ITC/with 0.05% DMSO (as a control) for the next 48 h. After 24 h, the medium was exchanged with fresh one in all wells. Zebrafish embryos xenotransplanted with cancer cell lines were anesthetized with 0.02% tricaine and imaged by fluorescence microscope just before (0 h) and 48 h from starting the treatment. The size of the tumor area was measured as a fluorescence intensity with ImageJ software (NIH, Bethesda, MD, USA) at the beginning of the experiment (0 h) and after 48 h after treatment. The tumor mass was calculated as a percentage-the average tumor size after 48 h compared to starting 0 h point. At the end of the experiment, confocal microscopy was used–we wanted to avoid monitoring the embryos during the experiment to minimize the laser exposition time. For confocal microscopy analysis the zebrafish larvae were anesthetized with 0.02% tricaine. Then they were examined with a LSM 700 confocal laser scanning microscope (Zeiss, Germany). The eGFP and DiI were excited by solid-state lasers (488 nm and 555 nm, respectively). For tumor-focused images 20× objective was used and z-stacks at 2-micron intervals were taken using ZEN software (Zeiss, Germany) for each channel. Stacks of images were compiled to produce maximum intensity projection images with ZEN software (Zeiss, Germany). Additionally, each optical section was analyzed separately, section by section, to detect even small details.

### 2.7. Statistical Analysis

We used GraphPad, Prism 6.00 for Windows, Graph Pad Software, San Diego, CA, USA). Data are reported as mean ± SD. Data were analyzed using a non-parametric U-Mann–Whitney test (GraphPad, Prism 6.00 for Windows, Graf Pad software, San Diego, CA, USA). The *p*-value < 0.05 was considered statistically significant with * *p* < 0.05, ** *p*  <  0.01 and *** *p*  <  0.001.

## 3. Results

### 3.1. Structurally Diverse Isothiocyanates in 1–10 µM Concentration Rate Showed a Different Toxic Effect in Zebrafish Embryos

To investigate the developmental toxicity of ITCs LJ-NCS-1–5 (1–10 µM) on 6 hpf and 48 hpf zebrafish embryos, the mortality, malformation and morphological abnormalities rates were recorded 24, 48, and 72 h after treatments. ITCs concentrations were set up based on the previous experiments.

Here, we aimed to find the non-toxic ITC/-s during 72 h of treatment for younger (6 hpf) and older (48 hpf) zebrafish embryos.

First, we assessed the ITCs/LJ-NCS-1–5 toxicity for a younger/ more sensitive embryos, and we found that the highest concentration (10 µM) of all substances was lethal to them (Figure 5). From all tested ITCs, the LJ-NCS-3 showed the strongest toxicity and 100% mortality rate in all doses. Next, 2–5 µM of LJ-NCS-1 was harmful to all tested organisms, whereas 5 µM LJ-NCS-5 induced yolk malformations in 35% *p* < 0.05) of embryos. Zebrafish embryos were not affected by 1–1.5 µM LJ-NCS-1, 1–5µM LJ-NCS-2 and LJ-NCS-4 and 1–4 µM LJ-NCS-5 at this time point.

After 48 h the 1–1.5 µM LJ-NCS-1, 1–4 µM LJ-NCS-2, 1–1.5 µM LJ-NCS-4 and 1µM LJ-NCS-5 concentrations showed no significant increase in the death rate and no deformations were detected in all cases (Figure 6). At this time point, 5 µM LJ-NCS-2, 2–5 µM LJ-NCS-4 and 1.5–5 µM LJ-NCS-5 were lethal to all embryos.

After 72 h the hatching rate was significantly decreased in the groups treated with 1.5 µM LJ-NCS-1, 4 µM LJ-NCS-2 and 1.5 µM LJ-NCS-4 (~25–30% of unhatched embryos; *p* < 0.05) compared with the control group (Figure 7). A total of 1 µM LJ-NCS-5 was lethal after 72 h of treatment. Totals of 1 and 1.5 µM of LJ-NCS-1 did not significantly decrease viability (94% and 86% viable embryos, respectively); however, both led to immobile embryos. Then, the majority (~83%; *p* < 0.001) of viable embryos exposed to 1 µM of LJ-NCS-4 demonstrated morphological malformations. Only 1–3 µM LJ-NCS-2 exposure did not induce any toxic effect in zebrafish embryos with a high viability ratio, being ~95–98%.

### 3.2. Phosphonate Analog of Sulforaphane as the Safest Treatment for Zebrafish Embryos at Different Stages of Development

Based on our observations using 6 hpf zebrafish embryos treated with ITCs, for further analysis with 48 hpf embryos, we tested a narrower doses range: 1–3 µM. In consequence, the 24 h treatment with 2–3 µM LJ-NCS-3 and 3 µM LJ-NCS-4–5 demonstrated a significant (2 µM LJ-NCS-3 and 3 µM LJ-NCS-4, *p* < 0.01; 3 µM LJ-NCS-3, *p* = 0.0043; 3 µM LJ-NCS-5, *p* = 0.0003) reduction of viable embryos and strong morphological malformations (especially of axis/in the tail region) in most (~86–92%; *p* < 0.001) analyzed embryos (Figure 8).

The 48 h time point demonstrated the lethality in the case of 1–3 µM LJ-NCS-3, 2–3 µM LJ-NCS-4 and -5 (Figure 9). The 3 µM LJ-NCS-1, 1µM LJ-NCS-4 and LJ-NCS-5 exposure revealed a significant (3 µM LJ-NCS-1, *p* = 0.0079; 1µM LJ-NCS-4, *p* = 0.0006; LJ-NCS-5, *p* = 0.025) decrease of viability and increase of fishtail deformations in ~55% of tested embryos. The exposure to 1–2 µM LJ-NCS-1 and 1–3 µM LJ-NCS-2 did not affect zebrafish embryo survival and body shape.

Next, after 72 h, 1–2 µM LJ-NCS-1 did not decrease the viability of tested embryos; however, the subjects were motionless; the 3 µM dose caused substantial survival reduction and fishtail deformations in ~70% of cases (*p* = 0.006) (Figure 10A). LJ-NCS-4 killed all tested subjects. LJ-NCS-5 at the minimal dose remarkably (*p* = 0.012) decreased the survival of zebrafish embryos; however, the viable ones were in good condition. Finally, at all time points, the LJ-NCS-2/P-ITC did not reduce the survival and its presence did not lead to any morphological malformations. Additionally, we checked the highest tested dose of P-ITC (3 µM) in the context of vascularization (main and enteral vessels), and we did not observe any difference between treated and control embryos after 24, 48 and 72 h from the starting point (48 hpf; Figure 10B).

Based on all results using 6 and 48 hpf embryos, we selected P-ITC (3 µM) for further studies using human cancer xenograft model in zebrafish embryos.

### 3.3. Phosphonate Analog of Sulforaphane Significantly Reduces the Human Cancer Cells Growth in Zebrafish Xenograft Models

The zebrafish model has recently emerged as a new system to study cancer progression which allows assessing cancer cell invasion over a short timescale. Here we performed the experiment based on various cancer cell lines injected into different organs, which aimed to show the universal effect of the tested compound. In the assay, U87, HeLa and MDA-MB-231 cancer cells were labeled with a red fluorescent Dil dye and injected into the hindbrain ventricle (Figure 11A), yolk sac region (Figure 11B) and Cuvier duct (transported to caudal hematopoietic tissue via blood vein) Figure 11C) of 2-day old embryos. After 24 h we selected human cancer cell-positive embryos and performed 48 h treatment with 3 µM P-ITC; control with 0.05% DMSO. The P-ITC exposure of zebrafish xenografts resulted in a significant decreases in glioblastoma (*p* = 0.0022), cervical (*p* = 0.028) and breast cancer (*p* = 0.026) growth compared to the non-treated control. To validate the tumor growth, we measured the size of the fluorescent rim in the brain, yolk, and caudal hematopoietic tissue in the same individuals (seeded separately in a 48-well plate) at the beginning of the study (0 h) and 48 h after treatment. Thus, we calculated the tumor size after 48 h (in percent) after treatment compared to the starting point of the experiment (0 h—the start of the treatment). At the end of the experiment, we analyzed embryos under a confocal microscope (Figure 11D). We did not observe metastatic sites during experiments and tumor cells occurred only on the site of injection.

## 4. Discussion

The evidence emerging from many in vitro and in vivo studies has revealed that ITCs show promising anti-tumoral potential [34]. The ability of ITCs to inhibit tumorigenesis depended on the target tissues, animal species, specific carcinogen employed and their structure [4,34]. For example, a longer alkyl carbon chain of arylalkyl ITCs is directly proportional to stronger chemopreventive effect in NNK-induced mouse lung carcinogenesis [35].

The mechanisms of cytotoxic and cytostatic effects of ITC include acceleration of apoptosis, blocking the cell cycle progression and angiogenesis [36,37]. Regulation of apoptosis by ITC is accomplished through the mitochondrial release of cytochrome c, Bcl-2 family regulation, MAPK signaling and subsequent activation of caspase −3 and −9 [38,39]. Cell cycle arrest caused by ITC occurs mainly in the G2/M phase. As a result, G2-phase regulators, including cyclin B1, cell division cycle (Cdc) 2 and Cdc25C, are down-regulated or inhibited and tubulin polymerization and mitotic spindle assembly are disrupted [40,41].

The previous study performed by Gajda D. et al., 2017 on P-ITC fully characterized its chemical properties and showed its anti-proliferative activity in vitro in cells derived from the human colon, lung, mammary gland and uterus carcinomas. P-ITC treatment led to the arrest of cell cycle and effectively induced cell death. Furthermore, P-ITC anti-tumor activity was confirmed in vivo using murine mammary gland carcinoma 4T1 model, in which it reduced tumor mass after compound administration [8].

In the present study, we assessed the toxicity of structurally different ITCs with exposure of zebrafish embryos starting 6 and 48 hpf and monitoring the effect 24, 48 and 72 h after exposure. These experiments validated the toxicity of ITCs on the varied stages of zebrafish embryos development, including the early–6 hpf and older 48 hpf embryos. The use of the young embryos (6 hpf) in the experiments allowed for determination if ITCs are harmful to the early stage of embryogenesis, while the tests on 48 hpf embryos provided the necessary basis for the xenograft transplantation experiments. Thus, the five tested ITCs showed the dose and structure-dependent manner of toxicity. LJ-NCS-3 presented the highest toxicity with lethality or induction of embryos body deformations at all time points of treatments for most tested doses. We found that 1–3 µM LJ-NCS-1 after 72 h was toxic for both 6 and 48 hpf embryos and impaired development, movement and/or morphology; additionally, every higher dose was lethal for the organisms. Next, 1–1.5 µM of LJ-NCS-4 allowed for survival of embryos at the end of exposure; however, the treatment caused 6 hpf zebrafish embryo axis and tail defects and led to delayed hatching. For 48 hpf embryos, LJ-NCS-4 was harmful in all tested doses. Then, 1 µM LJ-NCS-5 exposure left 48 hpf embryos with proper morphology; however, the survival rate was significantly decreased. Moreover, this substance was toxic for the younger zebrafish embryos. Finally, LJ-NCS-2 (P-ITC) showed good outcomes for 6 and 48 hpf embryos and 1–3 µM did not affect morphology, development, movement and vascularization.

To summarize, this set of experiments allowed us to select LJ-NCS-2/P-ITC as the non-toxic ITC treatment for zebrafish embryos in various stages of development.

Embryonic zebrafish is increasingly used as a toxicological model to conduct in vivo developmental toxicity assays [42]. Zebrafish features genetic homology and similarity of molecular pathways to humans, robust phenotypes, the possibility of high throughput genetic and chemical screening [42,43,44]. The toxicity testing using the fish embryos can be carried out more quickly than the standard test with mammals, uses much less space, costs, efforts and may uncover additional sub-lethal effects [45].

Recently, zebrafish has been established as a vertebrate model system useful in studying human diseases [46] with outcomes showing high probability of relevance to mammals, including humans [47]. The rapid development of zebrafish embryos and their permeability render this model suitable for screening novel molecules/substances/pharmaceutics and studies of the practical effects [48]. Furthermore, extending the usefulness of these vertebrates, current reports have shown the feasibility of transplanting cancer cells into zebrafish embryos to study tumor proliferation and growth mechanisms [49]. The zebrafish embryo lacks anti-tumor immunity and will not reject transplanted human tumor cells, partly because the T-cell receptor gene is not expressed in extrathymic sites before 9 days post-fertilization [50].

Based on that, we checked the anti-tumorigenic activity of the selected agent−3 µM P-ITC, against three highly malignant cell lines derived from glioblastoma (U87), cervical (HeLa) and breast (MDA-MB-231) carcinomas. Initially, by a standard scratch assay, we established that 3 µM P-ITC can inhibit the motility of cancer cells (data not shown).

Herein, the red fluorescence-labeled tumor cells were injected into the Tg(fli1:eGFP) transgenic zebrafish line showing GFP expression in vasculature. The transplantation was performed injecting three cell lines into different places to show that P-ITC can affect successfully various tissues and affect the growth of various tumor types; following U87 cancers were injected to the hindbrain ventricle, HeLa to the yolk sac region and MDA-MB-231 into the Cuvier duct through which cells migrated to the caudal hematopoietic tissue, by bloodstream. After 24 h we selected human cell-positive subjects and treated them 48 h with P-ITC. All three models showed a significant reduction of cancerogenic mass development compared to non-treated controls. The data confirmed the anti-tumor effect of P-ITC, which was previously studied in the murine breast cancer model and the results suggested that P-ITC has broader activity in the cancer tissue context and can actively inhibit various tumors growth. Additionally, as first we presented ITC effect using the zebrafish xenografts.

ITC functional moiety presented the ability to inhibit development of chemically induced cancer, oncogenic-driven tumor formation, and human tumor xenografts growth in rodents [51]. ITCs alone can suppress in vitro and in vivo growth of various cancer cells (such as non–small-cell lung cancer cells [52], human malignant astrocytoma cells [53], breast [54], ovarian [55] and HCT 116 human colon cancer cells [56]), but also sensitize them to cytostatic drugs used as standard in therapies [57,58].

In this context, in vivo tests demonstrated that benzyl isothiocyanate (B-ITC) suppresses glioblastoma GBM 8401/luc2 cell-induced tumor growth in athymic nude mice. By immunohistochemistry, the authors presented strong signals of caspase-3 and Bax in analyzed tissues [59]. Next, a series of artemisinin derivatives containing an ITC group were synthesized and tested with U87. Performed experiments showed in U87 cells activation of apoptotic pathways (by caspase-9 and -3 up-regulation), the autophagic mechanism (by LC3-II overexpression and silencing of p62) and significantly attenuated cell migration [60].

Anti-proliferative and pro-apoptotic activity of B-ITC was reported in in vitro studies using advanced ovarian cancer cells. The apoptotic mechanism was activated by caspase signaling and cleavage of PARP-1 [55]. Next, the accumulation of HeLa cells at the G2/M phase was observed 16 h after treatment with allyl isothiocyanate (A-ITC), benzyl or phenethyl isothiocyanate, accompanied by 41–79% of cell growth inhibition [61].

In breast cancer cells, the B-ITC affected carcinogen metabolism and signaling pathways involved in tumor progression and invasion. The authors proved that B-ITC inhibits breast cancer stem cell growth in association with suppressing the full-length receptor tyrosine kinase RON and its active form [51]. Phenethyl isothiocyanate induced apoptosis in HER2-expressing breast tumor-derived cells (MDA-MB-231 and MCF-7) in vitro and in vivo and enhanced the effects of doxorubicin [62]. On the other hand, A-ITC did not present inhibitory activity with MDA-MB-231 cells but stopped only the growth of MCF-7 [63].

## 5. Conclusions

The present study investigated the structurally different ITCs in vivo embryonic toxicity in zebrafish. According to obtained results, analyzing the effect of five ITCs, we selected P-ITC as the non-harmful treatment for the younger and older zebrafish embryos. These results indicate that structural modifications can significantly modulate the ITCs mode of action.

Next, our investigation demonstrated the high anti-cancer activity of P-ITC against various tumor cells (glioblastoma, breast and cervical carcinomas) accompanied by a high safety of treatment. In this context, our research demonstrated the applicability of the zebrafish embryo as a tool for the novel agent toxicity screening and in vivo model for the analysis of human tumor cell development. Future studies should focus on the molecular target analysis of the presented P-ITC effect and clinical trials.

## Figures and Tables

**Figure 1 cells-10-03219-f001:**
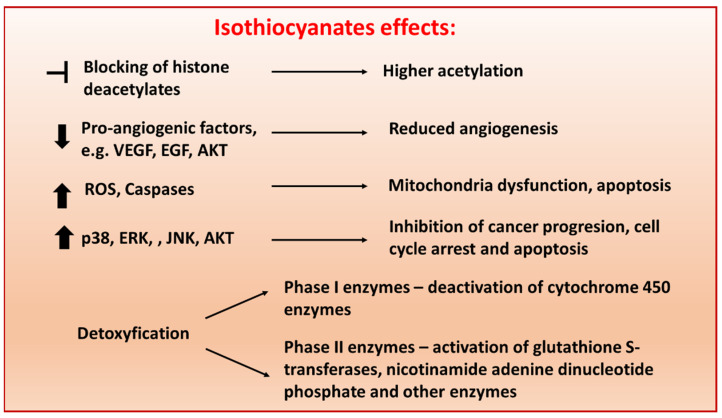
Schematic representation of molecular pathways targeted by isothiocyanates (ITCs) in cancer cells. ITCs exert their anti-tumor activities through various mechanisms, including (i) regulation of the epigenetic machinery, (ii) prevention of metastasis and angiogenesis, (iii) apoptosis via reactive oxygen species-initiated mitochondrial dysfunction (iv) inhibition of cell growth by causing cell cycle arrest and inducing cell death and (v) modulation of phase I and II enzymes [4]. The concentration in which ITCs present biological activity depends on chemical modifications and cell types.

**Figure 2 cells-10-03219-f002:**
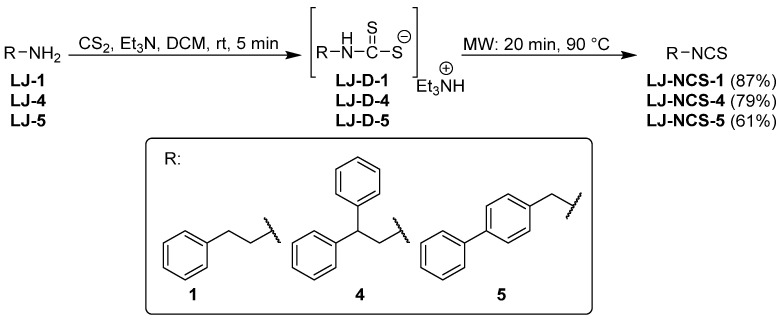
Microwave-assisted synthesis of LJ-NCS-1, LJ-NCS-4, and LJ-NCS-5.

**Figure 3 cells-10-03219-f003:**
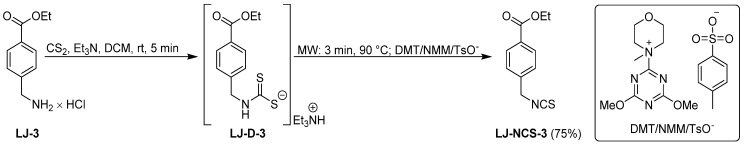
Microwave-assisted synthesis of LJ-NCS-3 using DMT/NMM/TsO^−^.

**Figure 4 cells-10-03219-f004:**
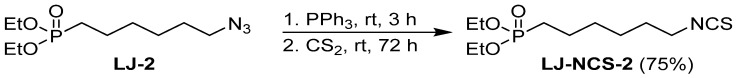
Synthesis of LJ-NCS-2 in Staudinger/aza–Wittig reaction.

**Figure 5 cells-10-03219-f005:**
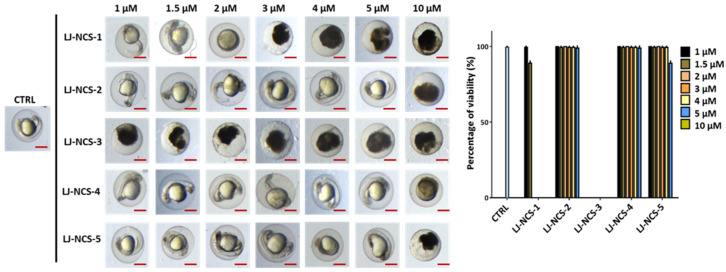
The effects of 24 h isothiocyanates treatment on morphology and mortality of 6 hpf zebrafish embryos at the experiment starting point. Representative images of embryos exposed to 1–10 µM of LJ-NCS-1–5 (n = 3 replicates, 50 embryos per replicate); bar scale = 450 µm. Percentage of viability-data are represented as mean ± SD.

**Figure 6 cells-10-03219-f006:**
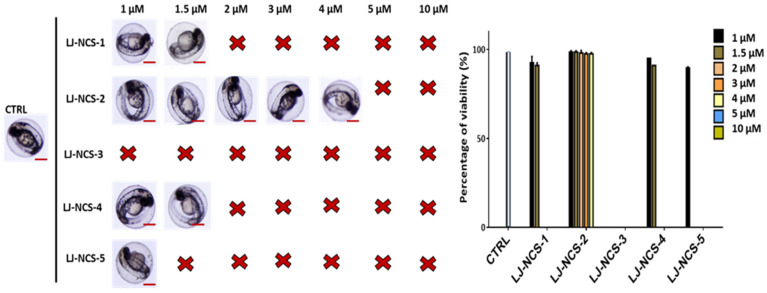
The effects of 48 h isothiocyanates treatment on morphology and mortality of 6 hpf zebrafish embryos at the experiment starting point. Representative images of embryos exposed to 1–10 µM of LJ-NCS-1–5 (n = 3 replicates, 50 embryos per replicate); bar scale = 450 µm Percentage of viability-data are represented as mean ± SD.

**Figure 7 cells-10-03219-f007:**
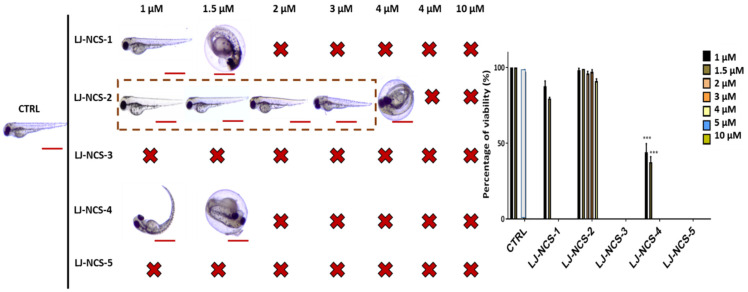
The effects of 72 h isothiocyanates treatment on morphology and mortality of 6 hpf zebrafish embryos at the experiment starting point. Representative images of embryos exposed to 1–10 µM of LJ-NCS-1–5 (n = 3 replicates, 50 embryos per replicate); bar scale = 950 µm. Percentage of viability-data are represented as mean ± SD; *** *p* <  0.001.

**Figure 8 cells-10-03219-f008:**
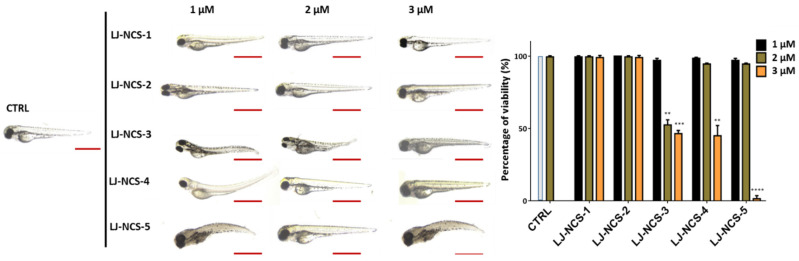
The effects of 24 h isothiocyanates treatment on morphology and mortality of 48 hpf zebrafish embryos at the experiment starting point. Representative images of embryos exposed to 1–3, of LJ-NCS-1–5 (n = 3 replicates, 50 embryos per replicate); bar scale = 950 µm. Percentage of viability-data are represented as mean ± SD; ** *p*  <  0.01 and *** *p*  <  0.001, **** *p*  <  0.0001.

**Figure 9 cells-10-03219-f009:**
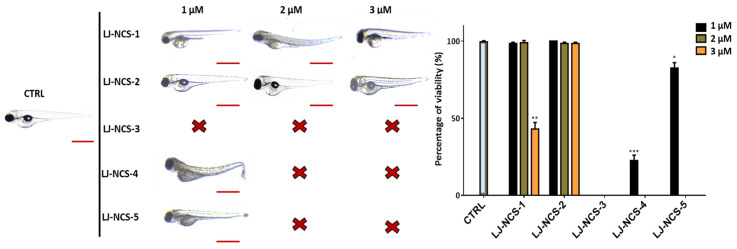
The effects of 48 h isothiocyanates treatment on morphology and mortality of 48 hpf zebrafish embryos at the experiment starting point. Representative images of embryos exposed to 1–3, of LJ-NCS-1–5 (n = 3 replicates, 50 embryos per replicate); bar scale = 950 µm. Percentage of viability-data are represented as mean ± SD; * *p*  <  0.05, ** *p*  <  0.01 and *** *p*  <  0.001.

**Figure 10 cells-10-03219-f010:**
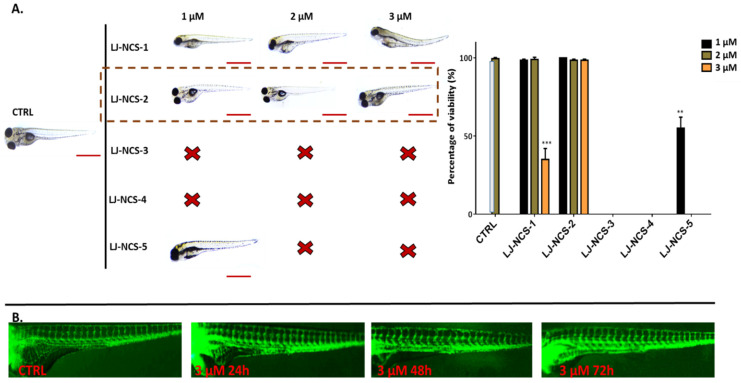
The effects of 72 h isothiocyanates treatment on morphology and mortality of 48 hpf zebrafish embryos at the experiment starting point. LJ-LNC-2 no effect on vascularization of zebrafish embryos. (**A**) Representative images of embryos exposed by 1–3, of LJ-NCS-1–5 (n = 3 replicates, 50 embryos per replicate); bar scale = 950 µm. Percentage of viability-data are represented as mean ± SD; ** *p*  <  0.01, *** *p*  <  0.001. (**B**) Representative picture (taken under fluorescence microscope) of blood vessels (green) of Tg(fli1:EGFP)y1 strain (48 hpf zebrafish embryos at the starting point) upon 3 µM LJ-NCS-2 treatment in 24, 48 and 72 h time points.

**Figure 11 cells-10-03219-f011:**
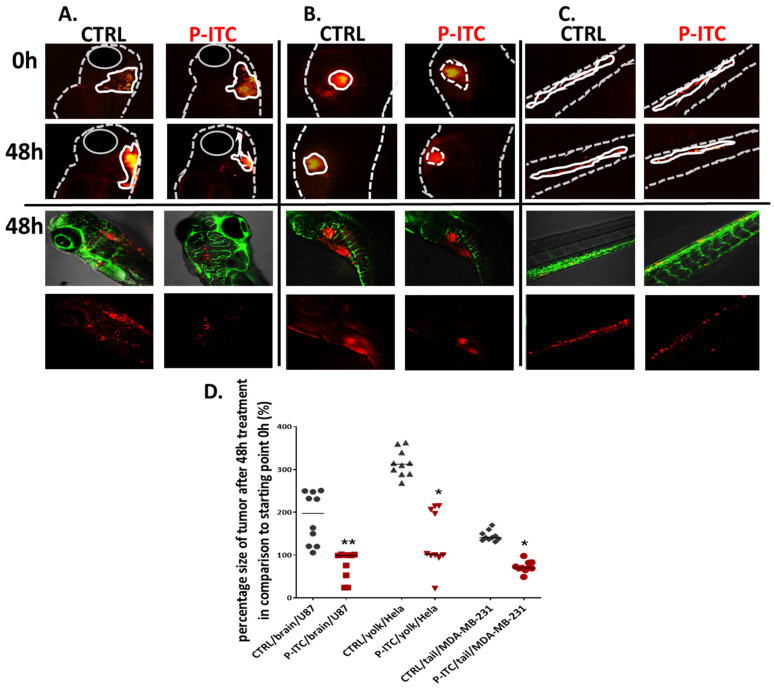
Phosphonate analog of sulforaphane (P-ITC) treatment inhibits tumor growth and fluorescence emitted from human glioma, cervical, and breast cancer cells grown in zebrafish embryos. The U87, HeLa and MDA-MB-231 cells were labeled with DiL (red fluorescence dye) and injected into the (**A**). hindbrain ventricle (U87), (**B**). yolk sac region (HeLa) and (**C**). in the caudal hematopoietic tissue (MDA-MB-231) of 48 h post-fertilization zebrafish embryos. After 24 h human cancer cell-positive embryos were selected and treated with 3 µM of P-ITC for 48 h. The images were taken under a fluorescence microscope and analyzed under a confocal microscope at the end of the experiment (lower panel of pictures; red–human fluorescence cells, green–blood vessels). The fluorescence emitted by the red tumor masses of each treatment group (n = 10) was measured by ImageJ and calculated in percentage. (**D**) The mass tumor growing of control and upon P-ITC treatment were measured at the beginning of the experiment (0 h; 24 h after human cells injection) and after 48 h of treatment (48 h). The presented percentage is the average tumor size after 48 h compared to starting 0 h point. Percentage of data are represented as mean ± SD; * *p*  <  0.05 and ** *p*  <  0.01.

**Table 1 cells-10-03219-t001:** Isothiocyanates tested in the experiments.

Symbol	Formula		Information
LJ-NCS-1	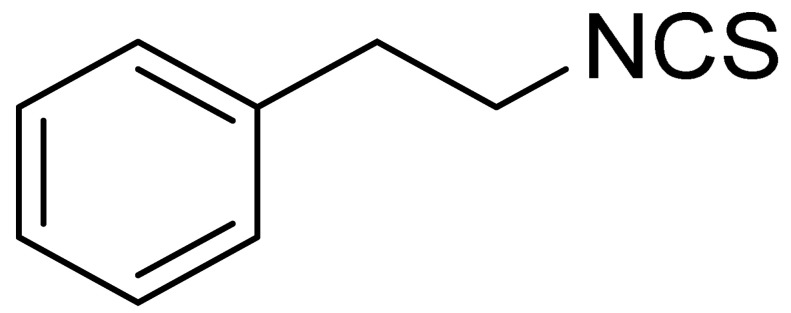	C_9_H_9_NS	Phenethyl isothiocyanate; natural isothiocyanate with high biological activity [25]
LJ-NCS-2	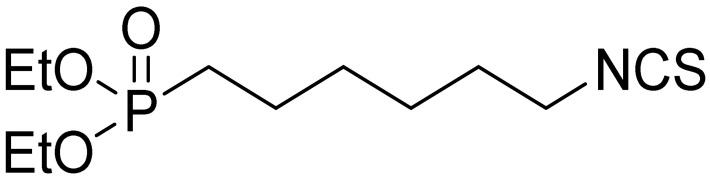	C_11_H_22_NO_3_PS	Phosphonate analogs of sulforaphane; synthetic isothiocyanate with anti-tumorigenic/anti-proliferative activity in vitro and in vivo [8]
LJ-NCS-3	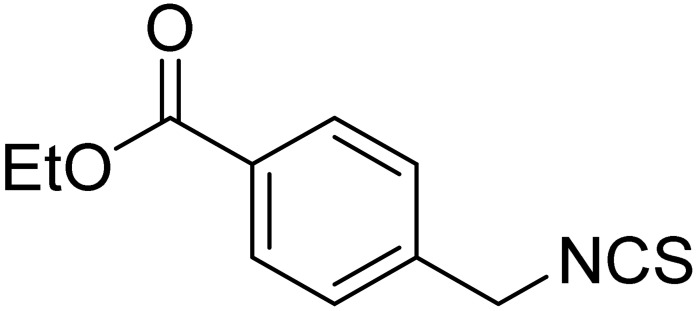	C_11_H_11_NO_2_S	A substance derived from 4-aminomethylbenzoic acid with hemostatic and antibacterial properties [26]
LJ-NCS-4	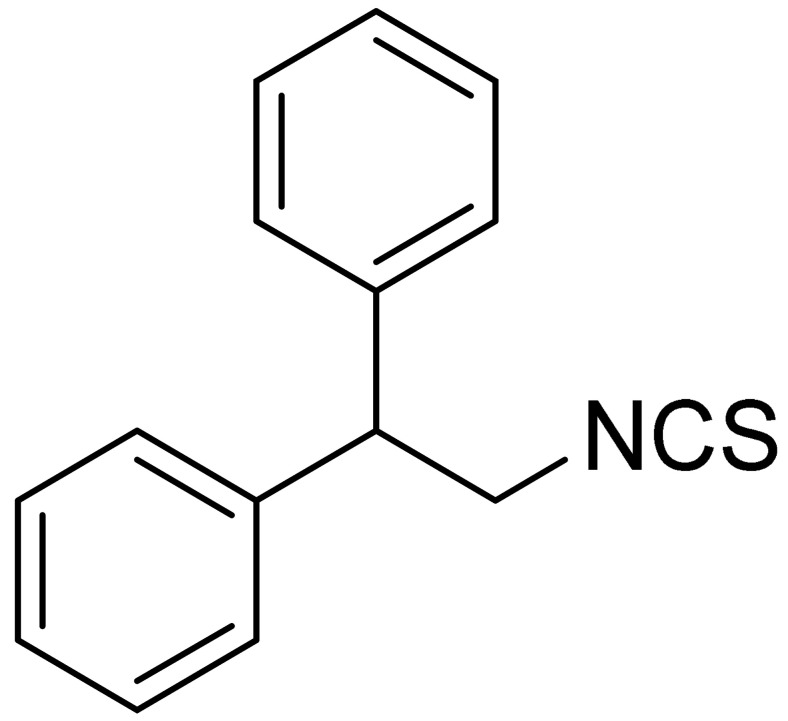	C_15_H_13_NS	A substance containing two aromatic rings with the potential to being anti- cancerogenic agent [27]
LJ-NCS-5	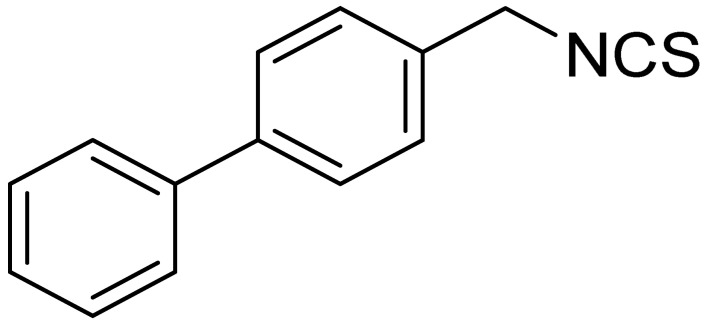	C_14_H_11_NS	A substance containing two aromatic rings with the potential to being anti- cancerogenic agent [27]

## Data Availability

The data presented in this study are available on request from the corresponding author. The data is not publicly available due to privacy restrictions.

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
