# Peer review of "The Anti-Tumoral Potential of Phosphonate Analog of Sulforaphane in Zebrafish Xenograft Model"

_cells, 2021, doi:10.3390/cells10113219_

Round 1

Reviewer 1 Report

This paper reports on the study of five synthetized isothiocyanates (ITCs) analogues to sulfaraphane in zebrafish embryo with the ultimate objective to study its anti-tumoral potential. From the five ITCs, the authors have tested one as non-toxic treatment at the concentration of 1-3 uM showing significant inhibition of tumor growth in different cell lines. In general, this study shows the anti-tumoral potential of this ITCs.

Despite being an interesting study with potential important scientific implications for the application of this compound in higher vertebrates, some revisions must be taken in consideration and some issues need to be addressed in order to improve the quality of this manuscript:

Abstract:

- Further information should be given on the methods used, exposure periods and concentrations tested in zebrafish that resulted in the selection of a single compound.

- The cell line used should be introduced.

- The conclusion should be focused on the results of this work and not cite other works already published.

Introduction:

- Regarding figure 1, at what concentrations are these effects observed? Furthermore, the legend states 4 mechanisms (form i to iv) but the figure shows 5. Please make the correspondence about the numbers and the different pathways.

- The objective should be reviewed to show the main objective which is not clear. Was it to synthetize new compounds? Test their toxicity using an alternative vertebrate model? Furthermore, have these ITCs been previously tested to know that they have potentially anti-cancerogenic properties?

- The last sentence of the introduction is more a conclusion of the work and should be changed to the appropriate section.

Material and Methods:

- Relative to the methods used for the preparation of each ITCs, if they have been previously described and characterized, the reviewer believes that there is no necessity to include so much information regarding the methods used to synthetize these compounds. Furthermore, have these compounds been chemically characterized to guarantee that the resulting product of the reaction was indeed the compound that the authors want? Which chemical techniques have been used to characterize the compounds?

- Regarding subtopic 2.4, the abstract state that embryos of 6 and 24 hpf were used but here 48 hpf are referred. Please review.

- Relative to the exposure, please clarify if both ages were exposed to 48 h.

- As for the methods used to assess embryotoxicity, please clarify the timepoints used. For instance, the hatching rate only makes sense to analyse from 48 hpf onwards.

- How was the impaired motor activity evaluated?

- Please include the methods used to dechorionated the embryos as well as the concentration of MS-222 used.

- Regarding xenograft exposure, what are the implications of exposing animals at 35 ºC knowing that rearing temperatures induce phenotypic sex changes?

- How was the confocal microscopy technique done?

- Why the Mann-Whitney test was used? Was data checked for normality? If so, please include further details on the statistical procedures.

Results:

- The results section should not contain references and should be focused on the results obtained.

- Please include the statistical tests information and the p-values for the comparisons made.

Reviewer 2 Report

In this study the authors assessed the toxicity of five different Isothiocyanates (ITCs) using the zebrafish embryo at 6hpf and 48hpf. The treatments durations ranged from 24-72h and the concentration of the compounds varied between 1-10uM. The authors used as a readout for toxicity: morphological defects, impaired movement, and viability. As a result, from the five ITCs only LJ-NCS-2 (P-ITC) had a non-toxic effect on the embryos and larvae. After assessing the concentration for the best survival rate, the authors injected 3 different human cell lines (human glioma, cervical, and breast cancer) in different regions of the zebrafish larvae at 48hpf. 24h post-injection, the xenografts were treated with LJ-NCS-2 (P-ITC) vs control and its effect on tumour reduction was quantified based on fluorescent signal.

The study is well performed but the final analysis of the xenografts could be significantly ameliorated and more thorough, increasing robustness and quality to the work. The English should also be revised.

Major points:

  1. The authors claim that LJ-NCS-2 (P-ITC) significantly reduced tumour size in glioblastoma, breast, and cervical zebrafish xenografts when compared with the controls. However, using the fluorescence intensity as a measurement to calculate tumour size is not the most precise one, especially when injecting in the yolk sac where many human cells tend to die and get cleared by phagocytes that take up the signal. To further confirm and increase the robustness and quality to the work I would suggest:
    • Sort the xenografts 24h post-injection by tumour size using the zebrafish eye as reference for example and proceed for control vs treatment.
    • The authors explain very well that ITCs “display anti-carcinogenic activity… including apoptosis, metastasis, vascularization, cell cycle, oxidative stress and epigenetic mechanisms” however, they only analysed tumour size. Those clearly should be considered, especially because the zebrafish model grants the potential to study all these cancer hallmarks, using immunofluorescence and confocal microscopy. I would suggest to the authors to perform in control vs drug:
      • Immunofluorescence (IF) to quantify mitotic figures (PHH3 staining);
      • IF for apoptosis (activated caspase3);
      • Analysis of angiogenic potential;
      • Always with DAPI to counterstain the nuclei
  1. Please clarify: “Thus, we calculated the size of the tumor after 48h (in percent) after treatment in comparison to the starting point of the experiment (0 h -the start of the treatment). Additionally, at the end of the experiment, we analyzed embryos under a confocal microscope and confirmed the reduced tumor size upon P-ITC exposure (Figure D). We did not observe a metastatic sites during experiments.”

It is not clear if graph D is from (calculated the size of the tumor after 48h (in percent) after treatment in comparison to the starting point of the experiment (0 h -  the start of the treatment)) – I guess using a stereoscope or if instead is what is explained in the text that graphD is from confocal – in that case you would only have the final time point and could not have the comparison with time zero…?

Minor points

  1. A native speaker should edit the manuscript before resubmission.

The manuscript has English problems that hamper the reading of the manuscript. Please revise.

  1. In the Abstract the authors say that they used “zebrafish embryos 6 and 24h post-fertilization”, this seems to be an error, since in all the paper (including Material and Methods) describes zebrafish larvae with 6 and 48
  2. Figure 1- I think the authors want to say acetylation not acetyation and Histone deacetylase
  • In line 310 (Figure D??) 11D?.
  • In line 310 : what do you mean with this? We did not observe a metastatic sites during experiments.- do not observe metastasis ?
  • Figures 7-11 zebrafish larvae should be positioned anterior (head) to the left and posterior (tail) to the right side. Dorsal up ventral down
  • In figure 11 there are several points to consider:
    1. zebrafish xenografts should be all positioned to the same side positioned anterior (head) to the left and posterior (tail) to the right side. Dorsal up ventral down
    2. Graphs should show each xenograft analysed not barrs – graphs with dots- where each dot is one xenograft
    3. There is a lot of autofluorescence in the images, if the authors intention was to show simultaneously the zebrafish larvae and the tumours, a brightfield image should be taken to show the larvae and another one with the red channel to show the tumour cells.
    4. The time is missing in the last panels of the figure- which we assume should be 48h?
    5. It seems that the last row should correspond to the fli:eGFP xenografts (previous row), although the images are not arranged in the same way as the fli:eGFP ones.
    6. The authors say that they injected breast cancer cells into the cuvier duct however, they are showing the cells in the Caudal Hematopoietic Tissue (CHT), according to the authors these cells are transferred to the tail by the blood flow, this should be more clear in the figure legend.

Round 2

Reviewer 2 Report

The manuscript has improved but I still think the charatherization of the phenotype by overall fluorescent is very poor...reducing the quality of the manuscript. Figure 11 could be much more stronger and does not make sense to do another manuscript with a carefull and thorough charatherization.

Also do not understand the in figure 11 why use IZO? and not the name of the P-ITC?